# ReLM: Leveraging Language Models for Enhanced Chemical Reaction Prediction

**Yaorui Shi**[‡]    **An Zhang**[§*]    **Enzhi Zhang**[¶]    **Zhiyuan Liu**[§]    **Xiang Wang**[‡†]

[‡]University of Science and Technology of China
[§]National University of Singapore, [¶]Hokkaido University
yaoruishi@gmail.com, anzhang@u.nus.edu,
enzhi.zhang.n6@elms.hokudai.ac.jp,
acharkq@gmail.com, xiangwang1223@gmail.com

## Abstract

Predicting chemical reactions, a fundamental challenge in chemistry, involves forecasting the resulting products from a given reaction process. Conventional techniques, notably those employing Graph Neural Networks (GNNs), are often limited by insufficient training data and their inability to utilize textual information, undermining their applicability in real-world applications. In this work, we propose **ReLM**, a novel framework that leverages the chemical knowledge encoded in language models (LMs) to assist GNNs, thereby enhancing the accuracy of real-world chemical reaction predictions. To further enhance the model's robustness and interpretability, we incorporate the confidence score strategy, enabling the LMs to self-assess the reliability of their predictions. Our experimental results demonstrate that ReLM improves the performance of state-of-the-art GNN-based methods across various chemical reaction datasets, especially in out-of-distribution settings. Codes are available at https://github.com/syr-cn/ReLM.

## 1 Introduction

Pre-trained language models (LMs) possess a vast reserve of knowledge, coupled with impressive capabilities for logical inference (Brown et al., 2020; OpenAI, 2023; Taylor et al., 2022; Chowdhery et al., 2022; Touvron et al., 2023; Chiang et al., 2023). These advantages render LMs useful for question-answering tasks, such as scientific inquiry and chemical inference (Bran et al., 2023; Shao et al., 2023). However, due to LMs' black-box natures, distinguishing whether the answers stem from their inherent knowledge or are arbitrarily generated poses a significant challenge. Furthermore, recent studies indicate that LMs struggle with the graph structures of molecules in chemistry-related tasks (Bran et al., 2023).

Recently, graph neural networks (GNNs) have been widely used to address chemical reaction problems due to their ability to handle complex graph structures inherent in chemical compounds (He et al., 2022; Somnath et al., 2021; Sacha et al., 2021). However, GNN-based methods often suffer from limited and biased training data, resulting in poor performance in real-world applications that involve diverse reaction mechanisms. This is exemplified by the unsatisfactory performance of LocalRetro (Chen and Jung, 2021) on Imidazo and NiCOlit (see results in Table 1), especially when encountering new reaction types absent in the training set. In addition, it is difficult for GNNs to effectively leverage the textual information (*e.g.,* reaction conditions) that could be derived from reaction descriptions. Identical reactants gas can yield different products depending on the catalyst used. An illustrative example is provided in Appendix C.1.

A crucial question arises: can we develop a chemical reaction framework that synergistically integrates the advantages of both GNNs and LMs? In this work, we propose **ReLM**, a novel framework designed to conduct chemical reaction prediction by utilizing both pre-trained LMs and GNNs. ReLM enhances the prediction accuracy in out-of-distribution (OOD) scenarios by utilizing graph structure understanding of GNNs and the natural language understanding capabilities of LMs. More specifically, ReLM employs GNNs to generate several high-probability candidate products. These products, along with appropriate in-context examples and descriptions of the reaction conditions, are then fed into the LMs to facilitate accurate chemical reaction prediction (as depicted in Figure 1). To further improve the robustness and interpretability of ReLM, we propose a prompting technique called the *confidence score strategy* (CSS). Harnessing

---

[*]Corresponding author
[†]Xiang Wang is also affiliated with Institute of Artificial Intelligence, Institute of Dataspace, Hefei Comprehensive National Science Center.

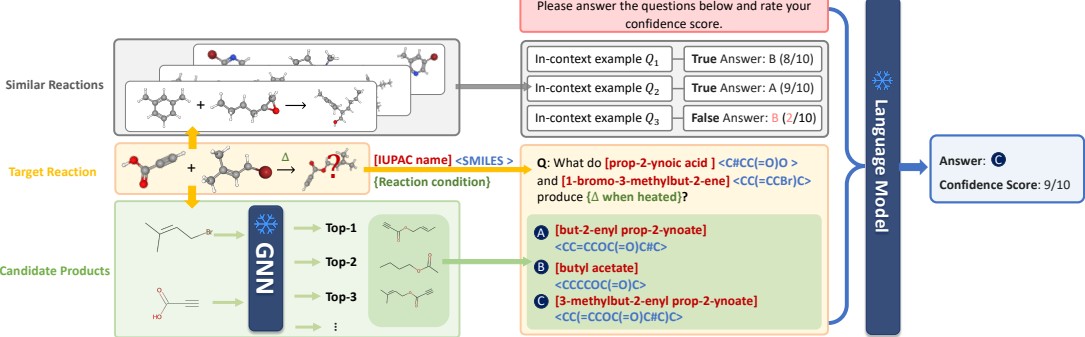

Figure 1: The overall framework of ReLM. The ReLM encompasses three key inputs for the language models: similar reactions (grey box), the target reaction along with its conditions (yellow box), and candidate products generated by GNNs (green box). Specifically, we select in-context examples with various confidence scores from the training samples that are nearest to the target reaction query. Additionally, we choose the $K$-candidate products with the highest likelihood as predicted by the GNN.

the intrinsic self-criticism capacity of LMs, CSS boosts the model's performance without involving significant computational costs (see Table 5). Extensive experiments on real-world Open Reaction Database (ORD) (Kearnes et al., 2021) demonstrate that ReLM achieves significant improvement over state-of-the-art GNN-based methods.

## 2 Background

**Chemical Reaction Analysis.** Recent studies utilize language models (LMs) for molecule analysis, relying solely on Simplified Molecular Input Line Entry System (SMILES) as input for molecular representation learning (Irwin et al., 2022; Chithrananda et al., 2020; Fabian et al., 2020). However, SMILES only provides one-dimensional linearizations of molecular structures, meaning that molecules with identical SMILES may exhibit entirely different properties (Wang et al., 2022). In contrast, GNN-based approaches take the structure of molecular graphs into consideration and have shown remarkable performance in in-distribution chemical reaction prediction tasks (Somnath et al., 2021; Dai et al., 2019).

A chemical reaction between reactant set $R = \{r_1, r_2, \cdots\}$ and product set $P = \{p_1, p_2, \cdots\}$ can be defined as:

$$r_1, r_2, \cdots \rightarrow p_1, p_2, \cdots. \quad (1)$$

Following the evaluation protocol established by Wang et al., we can consider reaction prediction as a task of ranking products within a predetermined product corpus $\mathcal{C} = \{P_1, P_2, \cdots\}$. This evaluation protocol ensures that $P_i \in \mathcal{C}$ holds for all reactions $R_i \rightarrow P_i$ in the tests.

Given a query with reactants $R$, the model is required to search across the product corpus to identify the most probable product. The likelihood between reactants and products is estimated using $L_2$ distance between their corresponding molecular embeddings:

$$D(R, P') = \| \sum_{r \in R} G(r|\theta) - \sum_{p \in P'} G(p|\theta) \|_2 \quad (2)$$

where $R$ is the given set of reactant molecules in a reaction, $P' \in \mathcal{C}$ is a product set from the corpus, and $G(\cdot|\theta)$ stands for the GNN model with parameter $\theta$.

**Prompting Strategies.** Instead of directly applying in-context learning to real-world tasks, some studies use prompting strategies for better performance (Shao et al., 2023; Bran et al., 2023). A commonly used strategy is instructing the language model to provide responses in a formalized format (*e.g.,* JSON or YAML). By using appropriate prompts, the thought process of an LM can be elicited using structured key-value pairs. Yang et al. develop the Multi-Query Ensemble Strategy (MES), which enhances the robustness of LMs. This strategy involves iteratively querying an LM with diverse in-context examples, conducting the inference process multiple times, and ultimately using a majority vote to determine the outcome.

## 3 Methodology

In this section, we propose ReLM, a framework combining graph neural networks (GNNs) and language models (LMs) for chemical reaction analysis. ReLM employs GNNs to generate answer candidates and in-context examples, then utilizes LMs for analysis in the form of multiple-choice

questions. We also propose the confidence score strategy, a generic prompting strategy to improve the reliability of LMs.

## 3.1 Context Generation

**Answer Candidates.** The prompt for language models can be formulated with input reactants $R$, reaction condition $c$, and a set of answer candidates $\hat{P}$ generated by GNN encoder.

The process of answer candidates generation can be succinctly described as data retrieval directed by a pre-trained GNN model.

Leveraging insights from the field of chemistry, chemical equations encapsulate the conservation of matter, charge, and various chemical properties. This implies a level of correspondence in the latent representations of the molecules on both sides of the equation. Therefore, for a specific set of reactants $R$, we employ the distance measurement function defined in Equation (2) to filter out those product sets $P_j \in \mathcal{C}$ that exhibit the highest similarity to $R$ in the latent space.

Formally, the answer candidates are generated by selecting the top-$K$ products with the highest similarity from the candidate pool $\mathcal{C}$:

$$\hat{P}_R = \text{TopK}_{P_j \in \mathcal{C}} -D(R, P_j) \qquad (3)$$

where $\hat{P}_R$ is the set of answer candidates for reactants $R$, and $D$ is the GNN-based similarity measurement defined in Equation (2).

**In-context Examples.** Besides the reaction information, the choice of in-context examples is also crucial for LM's few-shot learning performance. To acquire in-context examples that imply a reaction mechanism similar to the given reaction sample, we use the answer-aware example selection strategy proposed by Shao et al. (2023). Specifically, for a given test sample, we choose top-$N$ nearest neighbors from the training set based on the similarity of their reactants in the latent space, and use the $N$ training samples as in-context examples:

$$\mathcal{T} = \underset{i \in \{1,2,...,M\}}{\text{argTopN}} \frac{h_R^T h_{R_i'}}{\|h_R\|_2 \|h_{R_i'}\|_2} \qquad (4)$$

where $M$ is the size of the training set, $h_R = \sum_{r \in R} G(r|\theta)$ is the sum of all reactant representations, and $\mathcal{T}$ denotes the index set of the selected training samples. With the ground truth products $a_i$ of each training sample, the in-context examples are defined as:

$$\mathcal{E} = \{(R_i', c_i', a_i', \hat{P}_{R_i'})|i \in \mathcal{T}\} \qquad (5)$$

With reaction information $\{R, c, \hat{P}_R\}$ and in-context examples $\mathcal{E}$, the product prediction problem can be formulated as a single select multiple choice question. We input this question into the pre-trained language model, and the answer generated by the model is regarded as the output of our approach:

$$\hat{P} = \underset{P \in \hat{P}_R}{\text{argmax}} \; p_{\text{LM}}(P|\{R, c, \mathcal{E}\}) \qquad (6)$$

here $p_{\text{LM}}(A|\mathcal{P})$ is the probability distribution of answer $A$ in a language model given prompt $\mathcal{P}$ as input. To feed molecules into language models, we use both IUPAC names and SMILES string to represent molecules. In Appendix C.2, we present a detailed, step-by-step example of the context generation process.

## 3.2 Confidence Score Strategy

Besides prompting language models for chemical reaction prediction, we propose a universal prompting strategy named the confidence score strategy, which can be seamlessly transferred to any prompt-based language model application.

During inference, we ask the language model to report its confidence score (an integer number between 1 and 9) based on its understanding and familiarity with the given multi-choice question. To help the language model better understand how this score works, we also introduce the confidence score in the context prompt. For each in-context example, a random integer in $\{8, 9\}$ is chosen as the confidence score, which is then combined with the ground truth answer to form the prompt.

Nevertheless, the above random generation behavior can easily lead to a misunderstanding of the meaning of confidence scores by the language models, resulting in a degradation where the model only generates higher scores. Hence, we deliberately distort the answer of an in-context example to an incorrect one and assign it to a lower confidence score (*e.g.*, a random integer in $\{1, 2\}$). Thus, Equation 5 can be rewritten as:

$$\widetilde{\mathcal{E}} = \{(R_i', c_i', \widetilde{a_i}', \widetilde{s_i}', \hat{P}_{R_i'})|i \in \mathcal{T}\} \qquad (7)$$

where $\widetilde{a_i}'$ is the perturbed answer and $\widetilde{s_i}'$ is the corresponding confidence score. By asking for the confidence score, we let the language model

implicitly self-critique its answers during inference. In Appendix D, we verify that the confidence score strategy can be comprehended by language models.

### 3.3 Fine-grained Confidence Score Strategy

To achieve a more detailed analysis of confidence and enhance the language model's fine-grained interpretability, we also attempt to generate confidence scores for each answer candidate instead of a singular overall score. Subsequently, these candidates are re-ranked based on their respective confidence scores, selecting the one with the highest confidence as the definitive answer. We refer to this methodology as the *fine-grained confidence score strategy*. Please see more experimental results and analysis in Section 4.4 and Appendix B.4 .

## 4 Experimental Results

We aim to answer the following research questions:

**RQ1:** Can ReLM improve the reaction prediction capability of graph neural networks on both out-of-distribution and in-distribution data?

**RQ2:** Does the confidence strategy enhance the accuracy of ReLM?

**RQ3:** How does confidence score strategy influence the inference process of language models?

### 4.1 Experiment Setup

**Baselines.** Two popular GNN chemical reaction analysis methods, MolR (Wang et al., 2022) and LocalRetro (Chen and Jung, 2021), are selected as GNN backbones. For language models, we test GPT-3.5 (Brown et al., 2020) and Vicuna (Chiang et al., 2023). See more details in Appendix B.1.

**Datasets**. We utilize the USPTO dataset to train the GNN backbones. For evaluation, four datasets from the Open Reaction Database (ORD) (Kearnes et al., 2021) are utilized as the testbed for our experiments. The ORD contains diverse real-world reaction records from open-source projects and chemical literature. Notably, its utility for GNN-based chemical reaction predictions remains largely underexplored.

### 4.2 The OOD Capability of ReLM (RQ1)

**Motivation.** To evaluate the model's generalization ability, a comparison is made between our method and GNN baselines on the ORD dataset. Natural language descriptions of reactions, sourced from the ORD database, are utilized to enhance the quality of the prompts.

**Results.** In Table 1 and Table 5 (see Appendix B.3), we present the average accuracy of ReLM on the ORD dataset. ReLM improves the performance of GNN backbones across all test datasets. As ReLM utilizes the top-$K$ candidates provided by the GNN, the theoretical upper bound of our approach is constrained by the hit@$K$ accuracy of the GNN backbones. These upper bounds are also delineated in tables. To assess the stability of ReLM, we examine its accuracy under varying $K$-values and degrees of in-context randomness. Detailed results can be found in Appendices B.6 and B.7.

Apart from the evaluation on out-of-distribution datasets, we also run experiments under independent and identically distributed (i.i.d.) conditions. Refer to Appendix B.5 for more results.

### 4.3 Effectiveness of Confidence Score Strategy (RQ2)

**Motivation.** The Confidence Score Strategy proposed in this paper is a generalizable prompting method. To verify the effectiveness of this strategy, we conduct comparative experiments against other prompting strategies specifically tailored for multiple-choice questions. The baseline strategies include the Multiple-Ensemble Strategy (MES)(Yang et al., 2022), the JSON Strategy, and a control group that does not employ any strategy. MES necessitates multiple runs of the inference by the language model (10 times in our experiments). A majority vote is conducted based on these results, and the most frequently occurring answer is selected as the final outcome. By JSON strategy, we refer to the strategy that asks the language models to present their understanding of reaction precursors, reaction mechanisms, and its inferring process. Under this strategy, the language model is required to output all the above information along with its answer in a machine-readable format.

**Results.** In Table 2, we compare the overall accuracy (Acc), number of input tokens (#Token), and average inference time (Time/s) for each prompting strategy. Clearly, both the MES and JSON strategies enhance the model's performance, albeit with non-negligible time costs. Conversely, our proposed Confidence Score Strategy (CSS) prominently improves the accuracy without imposing a noticeable computational burden on the language model. See Appendix B.8 for comparison with more prompting strategies.

Table 1: Accuracy on out-of-distribution settings.

| | Imidazo | | NiCOlit | | Rexgen-30k | | Rexgen-40k | |
| | K=3 | K=4 | K=3 | K=4 | K=3 | K=4 | K=3 | K=4 |
|---|---|---|---|---|---|---|---|---|
| MolR (Wang et al., 2022) | 0.513 | 0.513 | 0.437 | 0.437 | 0.471 | 0.471 | 0.448 | 0.448 |
| **ReLM (MolR + Vicuna)** | $0.914^{+78.18\%}$ | $0.878^{+71.07\%}$ | $0.510^{+16.69\%}$ | $0.523^{+19.70\%}$ | $0.498^{+5.57\%}$ | $0.499^{+5.84\%}$ | $0.473^{+5.33\%}$ | $0.473^{+5.31\%}$ |
| **ReLM (MolR + GPT-3.5)** | $0.865^{+68.53\%}$ | $0.815^{+58.88\%}$ | $0.443^{+1.37\%}$ | $0.478^{+9.37\%}$ | $0.486^{+3.12\%}$ | $0.458^{-2.82\%}$ | $0.450^{+0.26\%}$ | $0.467^{+4.05\%}$ |
| Upper Bound | 0.945 | 0.979 | 0.640 | 0.679 | 0.584 | 0.611 | 0.562 | 0.586 |
| LocalRetro (Chen and Jung, 2021) | 0.023 | 0.023 | 0.086 | 0.086 | 0.279 | 0.279 | 0.245 | 0.245 |
| **ReLM (LocalRetro + Vicuna)** | $0.023^{+0.00\%}$ | $0.037^{+55.55\%}$ | $0.173^{+99.07\%}$ | $0.184^{+112.15\%}$ | $0.325^{+16.47\%}$ | $0.329^{+17.96\%}$ | $0.295^{+20.29\%}$ | $0.298^{+21.29\%}$ |
| **ReLM (LocalRetro + GPT-3.5)** | $0.031^{+33.33\%}$ | $0.037^{+55.55\%}$ | $0.224^{+157.71\%}$ | $0.254^{+192.22\%}$ | $0.348^{+24.64\%}$ | $0.364^{+30.37\%}$ | $0.308^{+25.44\%}$ | $0.316^{+28.70\%}$ |
| Upper Bound | 0.033 | 0.046 | 0.323 | 0.340 | 0.409 | 0.440 | 0.374 | 0.406 |

Table 2: Comparison of different prompting strategies within ReLM.

| | Rexgen-30k | | | Rexgen-40k | | |
| | Acc | #Token | Time/s | Acc | #Token | Time/s |
|---|---|---|---|---|---|---|
| MolR + Vicuna w/o strategy | 0.472 | 940.6 | 2.19 | 0.449 | 942.3 | 2.19 |
| MolR + Vicuna JSON | 0.456 | 1087.4 | 27.0 | 0.422 | 1091.6 | 27.5 |
| MolR + Vicuna MES | 0.490 | 9402.0 | 22.2 | 0.463 | 9412.3 | 22.4 |
| **MolR + Vicuna CSS** | 0.497 | 991.6 | 1.5 | 0.473 | 993.3 | 1.5 |
| MolR + GPT-3.5 w/o strategy | 0.464 | 956.6 | 2.09 | 0.420 | 966.4 | 1.63 |
| MolR + GPT-3.5 JSON | 0.456 | 1087.4 | 27.0 | 0.422 | 1091.6 | 27.5 |
| MolR + GPT-3.5 MES | 0.476 | 9564.1 | 18.9 | 0.426 | 9661.6 | 18.4 |
| **MolR + GPT-3.5 CSS** | 0.486 | 1007.6 | 1.6 | 0.450 | 1017.4 | 2.0 |

Table 3: Rank Correlation between ReLM's fine-grained confidence score and MolR's candidates ranking.

| | Imidazo | NiCOlit | rexgen-30k | rexgen-40k |
|---|---|---|---|---|
| $\rho$ | 0.363 | 0.359 | 0.346 | 0.347 |
| $\rho^+$ | 0.469 | 0.429 | 0.388 | 0.397 |
| $\rho^-$ | 0.243 | 0.227 | 0.199 | 0.174 |

It is noteworthy that ReLM achieves competitive or even superior results regardless of the prompting strategy employed for the language model. These results suggest that our reaction prediction method maintains a high level of robustness irrespective of the prompting techniques used.

### 4.4 Interpretability of Confidence Scores (RQ3)

**Motivations.** With more detailed confidence information, we aim to interpretably analyze the differences between language models and graph neural networks during decision-making processes, and investigate the reasons behind the performance enhancement brought by LMs. We employ Spearman's rank correlation coefficient as the metric to ascertain how the ranking produced by the LM correlates with the original rankings from the GNN model.

**Results.** For the evaluation of the fine-grained confidence score strategy mentioned in Section 3.3, we use MolR as the GNN backbone, Vicuna as the language model, and $K = 4$. In Table 3, the symbol $\rho$ denotes the rank correlation derived from ReLM and MolR across all test samples. On the other hand, $\rho^+$ and $\rho^-$ represent this correlation when MolR's predictions are correct and incorrect

respectively.

This correlation sheds light on the consistency between these two models in their decision-making processes. The overall rankings of the ReLM exhibit a positive correlation with the rankings of the MolR, implying that the ReLM concurs with most of the MolR's judgments.

Moreover, the ReLM concurs with the MolR when it's correct, and makes contradictory choices when it errs. This suggests that the ReLM is able to make informed judgments, diverging from the MolR when it believes the MolR is wrong. Additional results under the fine-grained confidence score strategy are in Appendix B.4. For further statistical analysis pertaining to confidence scores, refer to Appendix D.

## 5 Conclusion

Despite the great success of molecular structure understanding, today's GNN-based reaction prediction methods are still far from being able to do real-world reaction analysis. In this work, we proposed ReLM, an in-context prompting method that utilizes both language models and graph neural networks for chemical reaction prediction. Extensive experiments demonstrate the remarkable improvement of ReLM on various datasets indeed comes from the reasoning ability and self-criticism of LMs.

## Limitations

The limitations of ReLM are in two aspects, which will be addressed in future work. Firstly, ReLM focuses solely on chemical reaction prediction, neglecting other essential tasks in chemical reaction analysis such as retrosynthesis planning and yield prediction. Secondly, the performance gains achieved by ReLM are hindered by the drawbacks of backbone GNNs. The accuracy improvement is restricted by the hit@$K$ metric of GNN models, leading to only marginal advancements on specific datasets, as demonstrated in Table 1. We believe

our ReLM sheds light on chemical reaction prediction tasks and will inspire more work in this research line.

## Acknowledgements

This research is supported by the National Natural Science Foundation of China (9227010114), the University Synergy Innovation Program of Anhui Province (GXXT-2022-040), and the NExT Research Center.

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

## A  Datasets

The USPTO dataset used in Section 4 contains approximately 479k reaction samples, which are collected by Lowe (2012) and cleaned by Zheng et al. (2019).

We conduct our evaluation on four datasets extracted from Open Reaction Database (Kearnes et al., 2021). Considering there's an overlap between the two Rexgen datasets and USPTO, we exclude all reaction records in Rexgen-30k and Rexgen-40k that resulted in products already present in the training set. This operation ensures that the target products of test reactions have never been encountered during the training phase, imposing higher demands on the model's generalization ability.

The datasets, Imidazo and NiCOlit, encompass a wealth of textual descriptions of chemical reactions, which can be leveraged by language models for inferential purposes (refer to Section 4.2). The contents of the datasets utilized in this study are summarized in Table 4. Additionally, we provide a representation of some examples pertaining to reaction type and reaction condition information contained within these datasets, as depicted in Figure 2.

## B  Experiments

### B.1  Experiments Settings

**GNN models.** For MolR (Wang et al., 2022), we use the pre-trained checkpoints provided by its authors on their GitHub repository. The checkpoint model is trained on the USPTO dataset for 20 epochs with a batch size of 4096 and a learning rate of $1 \times 10^{-4}$. Specifically, we use a 2-layer TAG (Du et al., 2017) as the GNN encoder, as it has demonstrated superior performance in experiments of MolR. For LocalRetro (Chen and Jung, 2021), we train the model on the USPTO reaction prediction dataset. Following the experimental settings of Chen and Jung, we use a 6-layer GCN (Kipf and Welling, 2017) as the backbone GNN and train it for 50 epochs with a batch size of 16. The training process employs an Adam optimizer with a learning rate of $1 \times 10^{-4}$ and weight decay of $1 \times 10^{-6}$. During the Evaluation phase, we use 256 as batch size for both MolR and LocalRetro.

**Language models.** Training is not performed for the language model backbones due to the high training cost of LMs. We interact with GPT-3.5

through the OpenAI API, while for Vicuna, we employ the model released by its authors on HuggingFace (Wolf et al., 2019). It's worth noticing that due to the substantial financial costs associated with accessing GPT-3.5, we only use random subsets of size 500 of the test datasets when using GPT-3.5 as the backbone language model.

**Hyperparameters.** There are two important hyperparameters in our approach, the number of in-context examples $N$ (see Equation 4) and the number of answer candidates $K$ (see Equation 3). For the number of answer candidates, we try $K = 3, 4$, and $5$. For the number of in-context examples, we use a constant number $N = 3$ throughout our experiments.

### B.2  Implementation Details

For the speed test demonstrated in Table 2, we test all the involved methods on a single NVIDIA RTX A500 graphic card. For token count computation, we use tiktoken, a fast open-source byte pair encoding tokenizer that can be used with OpenAI's models. For fair comparisons, we also use this tokenizer to count the token usage of Vicuna (Chiang et al., 2023) in Section 4.3, even though it's not developed by OpenAI.

The training of LocalRetro requires atom-wise matching between reactants and products, but the training dataset used by us does not contain such data. We thus preprocess the training dataset with Rxnmapper (Schwaller et al., 2021) to meet this requirement.

Besides, as shown in Figure 1, we use IUPAC names along with SMILES to represent molecules in prompt design. However, the datasets only contain the SMILES expression of molecules. We leverage the database of PubChem (Kim et al., 2016) and open-source tool STOUT (Rajan et al., 2021) to perform the conversion from SMILES strings to IUPAC names.

### B.3  Ablation Study of Reaction Conditions.

Table 5 displays the results of ablation studies on descriptive reaction information. Incorporating reaction conditions (*e.g.,* reaction temperature and catalysts) and reaction type information in the prompts leads ReLM to achieve higher performance gains than compared to only reactant information.

Table 4: Statistics of the Four Datasets in the Open Reaction Database

| Name | Size | Description | Reaction SMARTS | Reaction Type | Reaction Condition |
|------|------|-------------|-----------------|---------------|--------------------|
| Imidazo | 384 | Three-component reaction approach towards diverse imidazopyridines reactions. | ✔ | ✔ | ✔ |
| NiCOlit | 1762 | Nickel-catalyzed cross-couplings reactions. | ✔ | ✔ | ✘ |
| Rexgen-30k | 7700 | Test data used by Rexgen (Coley et al., 2019). | ✔ | ✘ | ✘ |
| Rexgen-40k | 10235 | Validation data used by Rexgen (Coley et al., 2019). | ✔ | ✘ | ✘ |

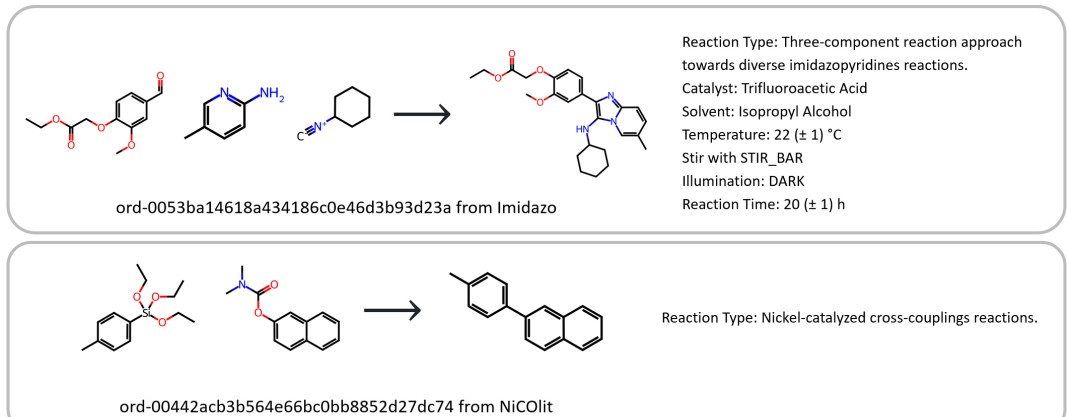

Figure 2: Examples of reaction type and condition in Imidazo and NiColit.

Table 5: Ablation study of textual reaction information.

|  | Imidazo | | NiCOlit | |
|--|---------|--|---------|--|
|  | GPT-3.5 | Vicuna | GPT-3.5 | Vicuna |
| MolR (Wang et al., 2022) | 0.513 | 0.513 | 0.437 | 0.437 |
| **ReLM (MolR)** | 0.812 | 0.765 | 0.412 | 0.521 |
| + reaction type | 0.822 | 0.903 | 0.478 | 0.523 |
| + reaction type and condition | 0.864 | 0.914 | - | - |

Table 6: Accuracies with multiple confidence strategy.

|  | Imidazo | NiCOlit | rexgen-30k | rexgen-40k |
|--|---------|---------|------------|------------|
| MolR | 0.513 | 0.437 | 0.471 | 0.449 |
| **ReLLM** | 0.870 | 0.507 | 0.449 | 0.421 |

### B.4 Accuracy of fine-grained Confidence Score Strategy.

In this section, we test the performance of the fine-grained confidence score strategy (introduced in Section 3.3). Table 6 displays the accuracies of both the ReLM and GNN models when using the multiple-CSS strategy. Although ReLM's performance does not surpass that of the original CSS strategy, it offers a greater degree of interpretability.

### B.5 Evaluation on In-distribution Dataset

The experiments in Section 4.2 demonstrate the powerful out-of-distribution performance of ReLM. We also conducted experiments under the independently and identically distributed (i.i.d.) setting on the test set of USPTO-479k. The results are shown in Table 7. This part of the experiment was conducted on the test set of USPTO-479k, using

Table 7: Evaluation on the test set of USPTO.

|  | MolR | LocalRetro |
|--|------|------------|
| MolR | 0.882 | 0.5663 |
| **ReLM (MolR+Vicuna)** | 0.871 | 0.6273 |
| Upper Bound | 0.9527 | 0.7583 |

Vicuna as the language model and MolR as the GNN backbone, with $K = 4$. Due to the extensive size of the USPTO dataset, we do not conduct experiments under all experimental settings.

In Table 7, it can be observed that GNN methods have solely achieved considerable performance in in-distribution scenarios, thus the performance enhancements provided by ReLM are limited in this circumstance. Furthermore, the lack of reaction type and condition information in the USPTO dataset also presents inference difficulties for the language model. Despite these challenges, ReLM still exhibited a certain capacity to select the correct answers from the options, without exhibiting significant performance deterioration.

### B.6 Effect of Different $K$-Values

Given that the accuracy upper bound of ReLM is determined by the number of answer candidates, it is necessary to evaluate the performance of the model under different $K$ levels. In this section, we conduct experiments across a wider range of $K$ values. The results are presented in Figure 3 using MolR as the GNN backbone and Vicuna as the language model. It can be observed in the table that ReLM's performance remains relatively stable

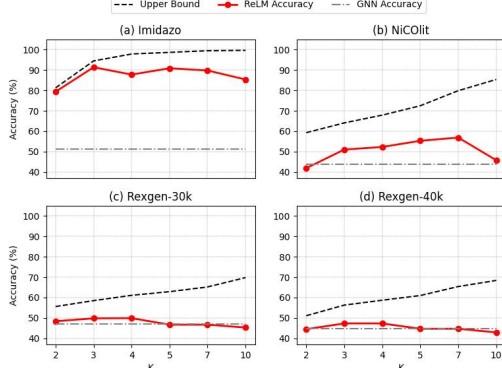

Figure 3: Accuracy under different $K$ values

Table 8: Accuracy over fixed and randomized in-context confidence scores.

|  | Imidazo | NiCOlit | rexgen-30k | rexgen-40k |
|---|---|---|---|---|
| GNN only (MolR) | 51.30% | 43.70% | 47.13% | 44.89% |
| Fixed: {1} and {9} | 91.93% | 53.94% | 48.65% | 45.62% |
| Randomized: {1,2} and {8,9} | 87.76% | 52.32% | 49.88% | 47.27% |
| Randomized: {1,2,3} and {7,8,9} | 91.41% | 53.78% | 48.51% | 45.52% |

as $K$ increases across a wide range of values (e.g. $K = 2 \sim 7$), and at some point, the performance gets better as $K$ increases. However, with very large $K$ values (e.g. $K \geq 10$), the accuracy of the model exhibited a noticeable decrease.

This drop in accuracy with extremely large $K$ values is likely because the candidates become overly saturated with implausible options, making it more difficult for the language model to discern the correct answer. Since there could be at most one ground truth candidate, increasing $K$ means adding more distracting and unlikely candidates. While the model is robust to these distractors up to a point, at some threshold of implausible options the task does become more challenging.

### B.7 Influence of In-context Randomness

In Section 3.1 we introduce the generation process of in-context examples. For each in-context example, we choose a random integer within $\{8, 9\}$ (or $\{1, 2\}$) as its confidence score. This intuitive random selection aims to faithfully mimic human behavior when answering the multiple-choice chemistry question. In this section, we demonstrate the efficacy of this strategy through comparative experiments.

In Table 8 we show the performance of ReLM under different levels of in-context randomness. The experiments are carried out using $K = 4$, Vicuna as the language model, and MolR as the GNN.

Table 9: ReLM with other strategies.

|  | Imidazo | NiCOlit | rexgen-30k | rexgen-40k |
|---|---|---|---|---|
| MolR | 0.513 | 0.437 | 0.471 | 0.449 |
| Zero-shot | 0.870 | 0.446 | 0.301 | 0.286 |
| Few-shot CoT | 0.781 | 0.305 | 0.267 | 0.244 |
| Zero-shot CoT | 0.844 | 0.415 | 0.318 | 0.302 |
| **CSS** | 0.878 | 0.523 | 0.499 | 0.473 |
| Upper Bound | 0.979 | 0.679 | 0.611 | 0.587 |

In the table, we can observe that using varying confidence scores causes slight fluctuations in experimental outcomes, though the degree varies by dataset. This further demonstrates that the effectiveness of our ReLM lies not in the details of prompt design. Instead, it stems from our main idea of amalgamation of the molecular modeling ability inherent to GNNs with the vast reaction knowledge of the language model.

### B.8 Comparison with More Prompting Strategies

In addition to the prompting strategies in Section 4.3, we also included Zero-shot, Few-shot CoT, and Zero-Shot CoT as baseline prompting methods. Table B.8 shows the experimental results of these methods.

We observe that all prompt designs offer some benefits. However, our original confidence score approach still outperforms these baselines. Specifically, as shown in the case studies in Section C.3, the language model's CoT analysis may not provide additional insightful information regarding the reaction mechanism. At this stage, incorporating additional analytical steps may introduce more molecular structures, exacerbating the language model's comprehension difficulties. Further step-by-step elucidation of the reaction process likely necessitates incorporating more domain knowledge of chemical reactions.

## C Case Studies

### C.1 Importance of Reaction Condition

Previous GNN-based reaction analysis models could only process molecular graphs and were unable to utilize the abundant information contained within chemical reaction conditions. However, many chemical reaction conditions exert profound influences on the reaction mechanisms, determining the direction of synthesis. In this section, we illustrate the importance of reaction conditions to reaction outcomes through an example from the Imidazo dataset.

In Figure 4, we report the answers from GNN and ChatGPT to two chemical reaction problems involving the same reactants. The only difference between these two problems is that problem 2 does not include reaction conditions and types. Question 1 is the target reaction given the correct catalyst, while Question 2 omits the important catalyst. For GNN, it incorrectly predicts 'B' as the answer for both questions. In contrast, the language model accurately identifies option 'C' for Question 1, which aligns with the ground truth. However, for Question 2, the language model predict a product that was not present in the candidate pool.

In reality, this organic reaction involves three components - an aldehyde, an amine, and an isocyanide. Without an added catalyst, the combination of these three reactants undergoes a Passerini reaction, which is a type of multi-component condensation. Nevertheless, in the presence of the trifluoroacetic acid catalyst, the reaction proceeds via an imidazopyridine formation mechanism. This leads to the major product being 'C', 3-[3-(cyclohexylamino)-6-methylimidazo[1,2-a]pyridin-2-yl]benzene-1,2-diol.

These case studies clearly demonstrate that identical reactants yield different products under different reaction conditions, further reinforcing our driving hypothesis: the incorporation of reaction type and condition boosts the predictive accuracy of ReLM.

### C.2 Illustration of Inferring Process

In Section 4.2, we mentioned that the accuracy upper limit of ReLM is determined by the hit@$K$ metric of the backbone GNN model. In this section, we furnish an illustrative example to elucidate our proposed methodology more comprehensively. Figure 5 provides a step-by-step demonstration of our method when $K = 4$. The in-context examples have been omitted for brevity.

### C.3 Examples of Prompting Strategies

In Section 4.3 we compare the confidence score strategy with other prompting strategies. In Figure 6, we use an example from the NiCOlit dataset to further demonstrate the difference between the involved prompting strategies.

### D Statistical Analysis

In this paper, we posit that language models possess the capability to comprehend confidence scores and assign corresponding scores based on their familiarity with the questions. In this section, we elucidate this claim by presenting the distribution of these scores across a large number of samples, that confidence scores can indicate ReLM's degree of comprehension of the queries.

We illustrate the distribution of confidence scores using the Rexgen-40 dataset, employing Vicuna + MolR as backbones. The results are presented in Figure 7. Confidence scores of ReLM exhibit significantly different distributions between correct and incorrect answers. For incorrect ReLM choices, the proportion of high confidence scores ($7 \sim 9$) decreased from $82.3\%$ to $51.2\%$, while that of low confidence scores ($1 \sim 3$) increased from $4.4\%$ to $22.7\%$.

Additionally, we designed experiments to compare the accuracy of the large model at different confidence levels. We divide the datasets into two parts based on the model's output confidence and calculate the accuracy separately. The results are shown in Table 10. The "High Conf." column represents the model's accuracy on the subset with higher confidence, and the "Low Conf." column represents the accuracy on the part with lower confidence. For all control groups, ReLM's accuracy on high-confidence samples is significantly higher than those with lower confidence ($p$-value $\ll 0.05$).

Table 10: Accuracy under high/low confidence levels. The "High Conf." column represents the accuracy of the model on the subset with higher confidence, and the "Low Conf." column represents the lower ones. ReLM's accuracy on high-confidence samples is significantly higher than those with lower confidence (p-value ≪ 0.05).

| | | Rexgen-30k | | Rexgen-40k | |
| --- | --- | --- | --- | --- | --- |
| | | k=3 | k=4 | k=3 | k=4 |
| MolR | | 0.471 | 0.471 | 0.448 | 0.448 |
| **ReLM (MolR + Vicuna)** | High Conf. | 0.500±0.235 | 0.502±0.234 | 0.477±0.229 | 0.478±0.229 |
| | Low Conf. | 0.361±0.300 | 0.329±0.305 | 0.330±0.291 | 0.288±0.293 |
| **ReLM (MolR + GPT-3.5)** | High Conf. | 0.525±0.201 | 0.525±0.257 | 0.499±0.160 | 0.510±0.198 |
| | Low Conf. | 0.438±0.268 | 0.333±0.313 | 0.400±0.245 | 0.440±0.254 |
| LocalRetro | | 0.279 | 0.279 | 0.245 | 0.245 |
| **ReLM (LocalRetro + Vicuna)** | High Conf. | 0.330±0.163 | 0.332±0.190 | 0.299±0.151 | 0.302±0.178 |
| | Low Conf. | 0.235±0.203 | 0.213±0.220 | 0.238±0.181 | 0.190±0.203 |
| **ReLM (LocalRetro + GPT-3.5)** | High Conf. | 0.368±0.130 | 0.405±0.132 | 0.321±0.117 | 0.337±0.150 |
| | Low Conf. | 0.248±0.203 | 0.270±0.219 | 0.200±0.181 | 0.262±0.193 |

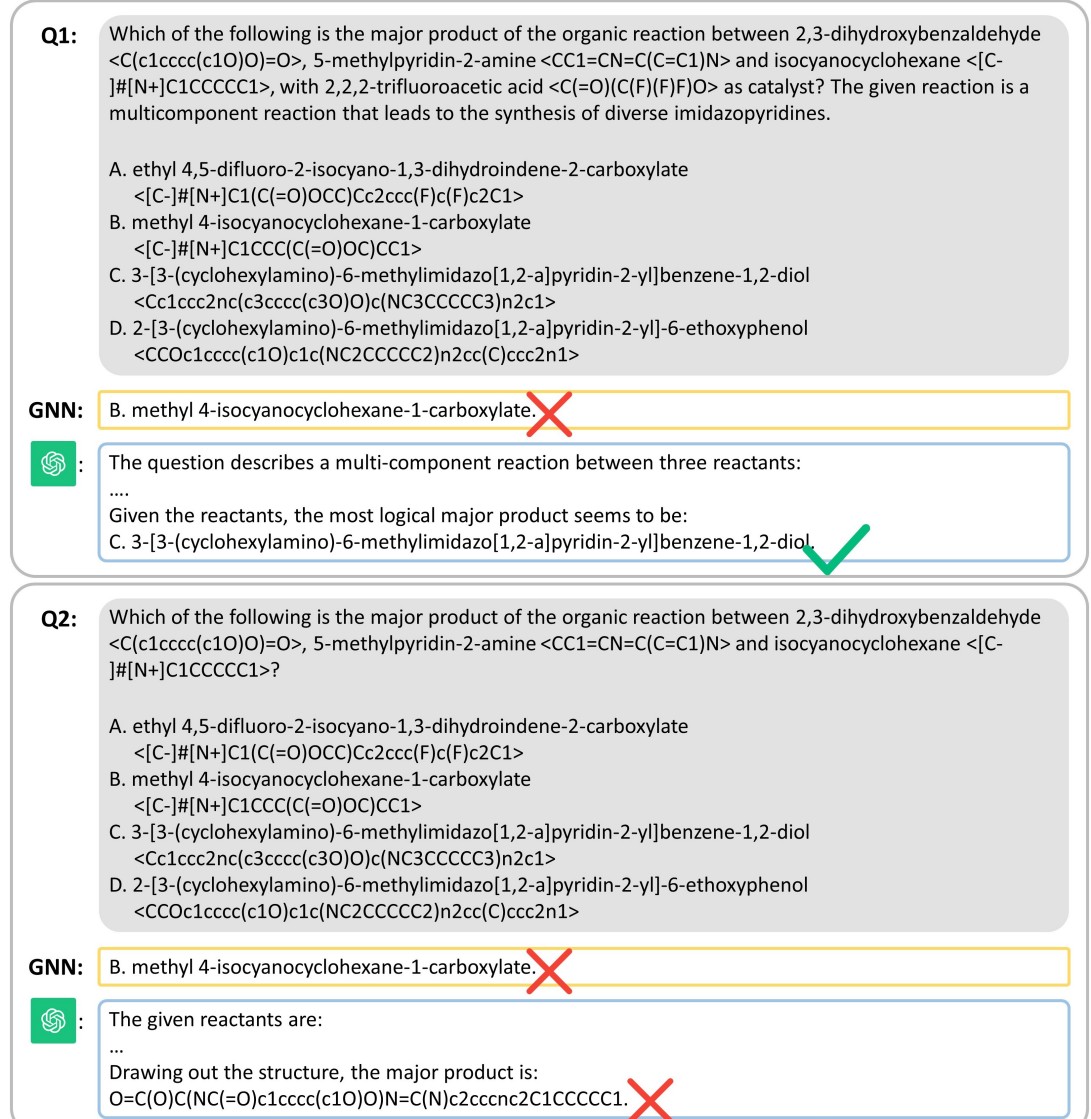

Figure 4: The answers from GNN and ChatGPT to two similar chemical reaction problems. The only difference between these two questions is the existence of reaction conditions. The GNN incorrectly predicts 'B' as the answer for both questions. In contrast, the language model accurately identifies 'C' as the answer for the first problem, but fails to predict the second question. These case studies demonstrate that identical reactants yield different products under different reaction conditions.

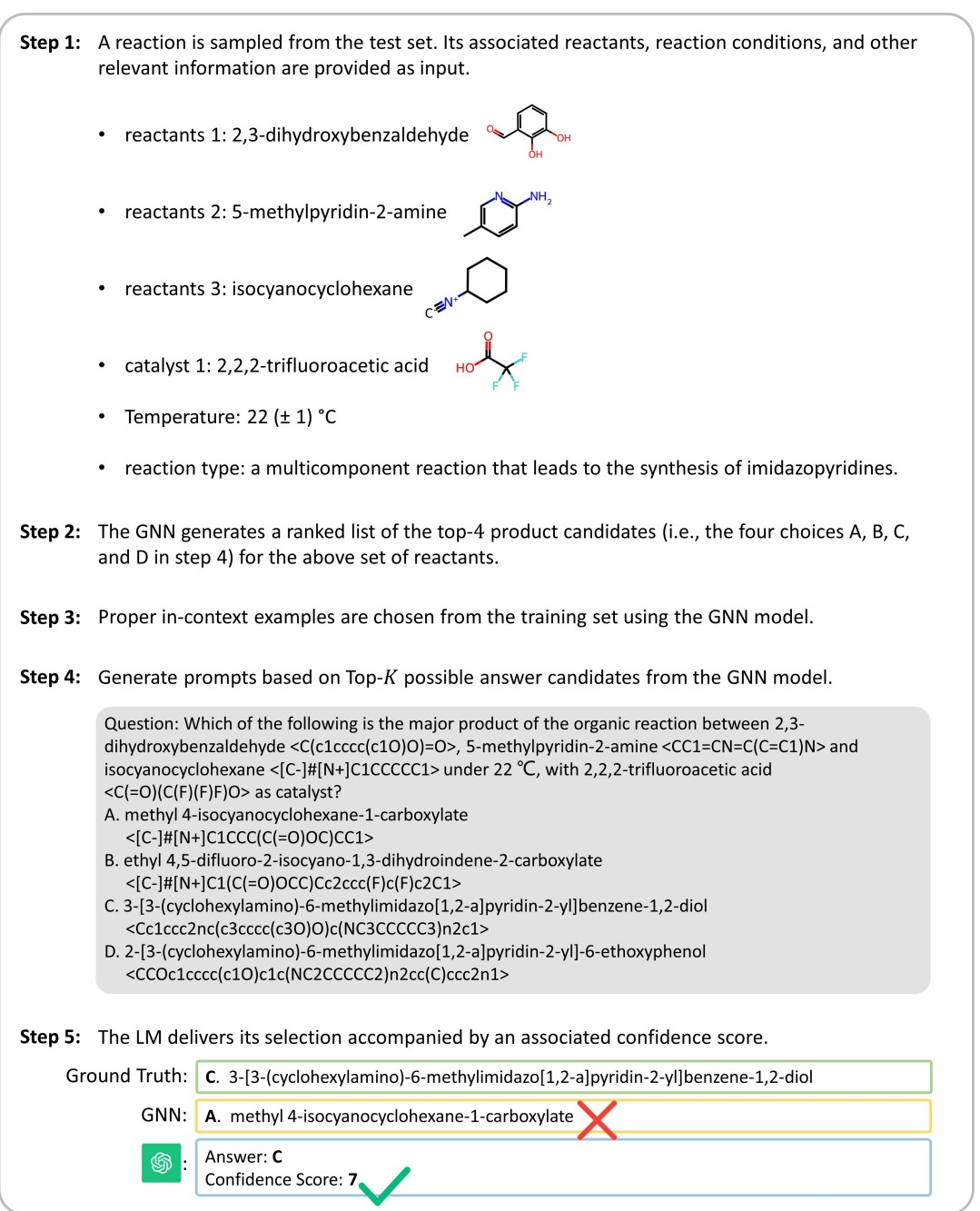

Figure 5: The step-by-step illustration of the inferring process. The test case is from the Imidazo dataset.

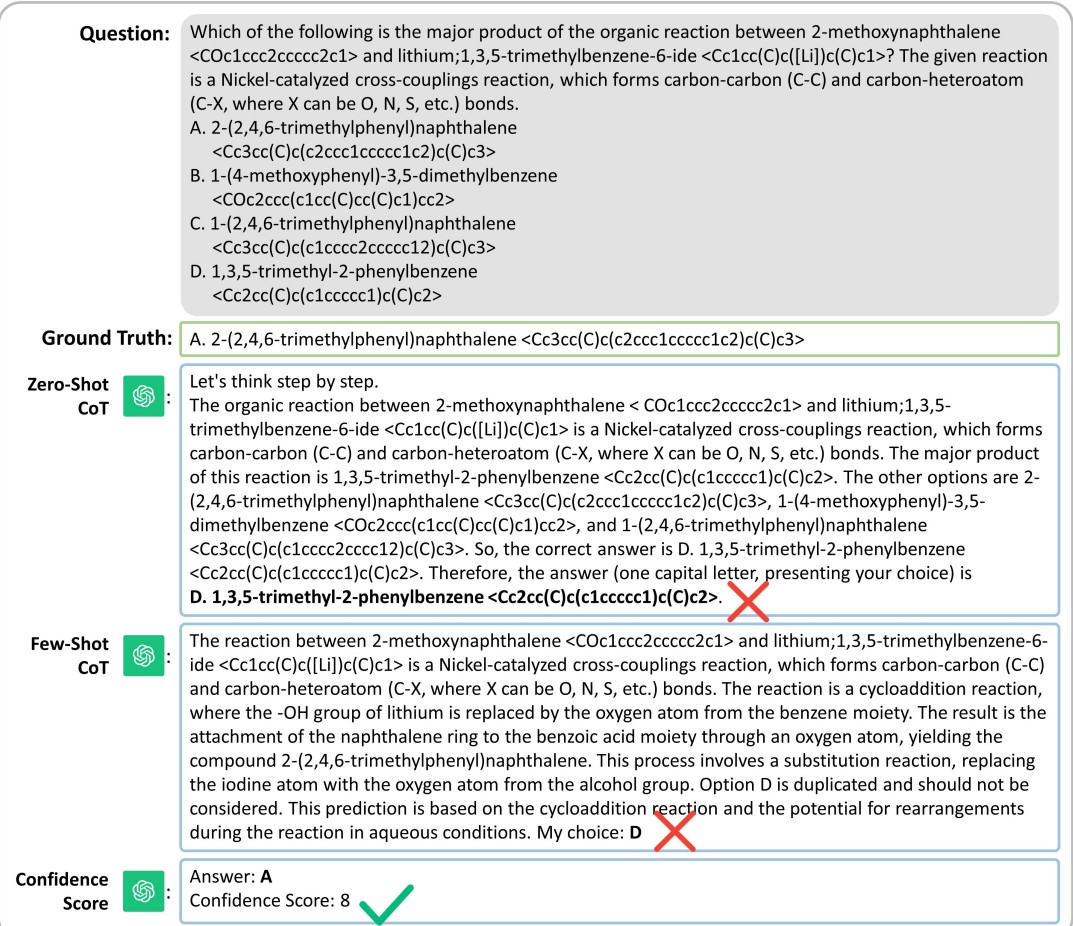

Figure 6: Illustration of different prompting strategies. The in-context examples in the few-shot CoT and Confidence Score strategies have been omitted for brevity.

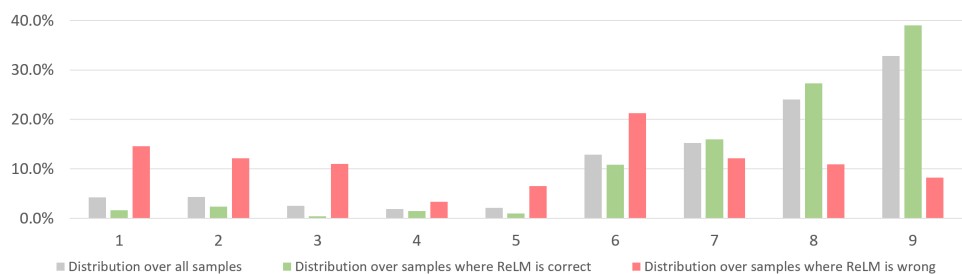

Figure 7: Distribution of Confidence Scores on Rexgen-40 dataset.