# OpenReview forum: "ReLM: Leveraging Language Models for Enhanced Chemical Reaction Prediction"
_EMNLP/2023/Conference — EMNLP 2023 Findings_

### Official Review · Reviewer_i7LA · 2023-07-25

**Soundness:** 4

**Excitement:**

3: Ambivalent: It has merits (e.g., it reports state-of-the-art results, the idea is nice), but there are key weaknesses (e.g., it describes incremental work), and it can significantly benefit from another round of revision. However, I won't object to accepting it if my co-reviewers champion it.

**Paper Topic And Main Contributions:**

This paper proposes a framework that tries to predict chemical reactions utilizing both GNN and recently emerged large language models named ReLM.
Given a test reaction, ReLM generates questions and answers from similar reactions together with a confidence score as in-context learning prompts and feeds them into LLM. GNN is used to provide candidate target pools. The author proposes a Confidence Score Strategy to generate a confidence score fir each question-answer pair to improve the effectiveness of LLMs.
In general, this is a well-written short paper with a simple method, lying more to the engineering side.

**Questions For The Authors:**

1.  Eqn (1): What is the size of the reactants R? Is it a list that contains all the reactants or a single reactant?  Is it fixed for a query? If there are multiple reactants, how do you leverage GNN to select top-K candidates? Do you pick top-K from only one of the reactants or all of them?

2. As discussed in weakness, for your confidence score strategy, why pick [1,2] and [8,9]? Why not just [1] and [9] or [1,2,3] and [7,8,9]? Wouldn’t the randomness bias the model?

3. Could you explain the difference between the baseline between the first row (method name) and last row (hit@K), and explain why hit@K is not enough to get results? How do you plan to extend this work to increase the generalizability and adaptability of ReLM?

Question importance: 3 > 2 > 1. I’d love to change my score based on the author's response.

**Reasons To Accept:**

This paper explores a simple way to combine knowledge from both LLM and GNN to enhance chemical prediction without the need to finetune the heavy LLM.

The improved results are shown on 2 GNN methods {MoIR, LocalRetro} x 2 LLM {GPT 3.5, Vicuna} testbed combinations.

Thorough ablation studies are shown to prove the effectiveness of the prompt design and confidence score strategy. I appreciate the writing as well as the experimental presentation of the works done.

Code availability is good.

**Reasons To Reject:**

1. The choice of baselines is a little bit confusing. If pre-training an in-house GNN is always required, and GNN hit@K is a theoretical upper bound, then why do we still need ReLM to introduce textual knowledge? This question is important as it is closely related to the use case of ReLM.

2. The novelty of methodology looks limited. The main idea contribution lies mostly on the prompt design part. The confidence score design is also a little bit weird, what’s the intuition to set the score choice as [1,2] and [8,9]? Why is randomly picking up a score reasonable?  Why do strict score assignments (e.g., 1 for bad and 9 for good) not work? Missing explanation to this further limits the novelty of the idea itself.

**Reproducibility:**

5: Could easily reproduce the results.

**Reviewer Confidence:**

5: Positive that my evaluation is correct. I read the paper very carefully and I am very familiar with related work.

---

> ### Author Rebuttal · Authors · 2023-08-29
>
> We sincerely thank you for your time and insightful comments. Your primary suggestions regarding our motivation and methodology have helped us improve the clarity of our work. To address your concerns, below we present the point-to-point responses.
>
> > **Comment 1: Significance of proposed method** — "The choice of baselines is a little bit confusing. If pre-training an in-house GNN is always required, and GNN hit@$K$ is a theoretical upper bound, then why do we still need ReLM to introduce textual knowledge? This question is important as it is closely related to the use case of ReLM."
>
> We appreciate you asking this important question. The hit@$K$ metric measures the performance of a ranking algorithm by checking whether the ground truth is within the Top-$K$ results. For example, hit@3 measures whether the correct result is in the Top-3 ranked items. Previous GNN approaches only considered the top rank 1 prediction as the final output, so model accuracy was solely determined by hit@1.
>
> In this paper, we propose analyzing the Top-$K$ results of the GNN model and using a language model to select the most plausible option as the final prediction. This enables potentially selecting the correct answer from the GNN's Top-$K$ choices (instead of Top-1). Therefore, we refer to the GNN's hit@$K$ metric as the theoretical upper bound on the accuracy of our model.
>
> To clarify, we prodive a step-by-step example of our methodology when $K=4$ below:
>
> - Step 1: A reaction is sampled from the test set. Its associated reactants, reaction conditions, and other relevant information are provided as input.
> 	- reactants 1: 2,3-dihydroxybenzaldehyde
> 	- reactants 2: 5-methylpyridin-2-amine
> 	- reactants 3: isocyanocyclohexane
> 	- catalyst 1: 2,2,2-trifluoroacetic acid
> 	- reaction type: a multicomponent reaction that leads to the synthesis of diverse imidazopyridines
> - Step 2: The GNN generates a ranked list of the top 4 product candidates (i.e., the four choices A, B, C, and D in step 4) for the above set of reactants.
> - Step 3: Proper in-context examples are chosen from the training set using the GNN model.
> - Step 4: Generate prompts based on Top-$K$ possible answer candidates from the GNN model. (The in-context examples are omitted for brevity.)
>
> ```
> Question: Which of the following is the major product of the organic reaction between 2,3-dihydroxybenzaldehyde<C(c1cccc(c1O)O)=O>, 5-methylpyridin-2-amine<CC1=CN=C(C=C1)N> and isocyanocyclohexane<[C-]#[N+]C1CCCCC1>, with 2,2,2-trifluoroacetic acid<C(=O)(C(F)(F)F)O> as catalyst?
> A. methyl 4-isocyanocyclohexane-1-carboxylate<[C-]#[N+]C1CCC(C(=O)OC)CC1>
> B. ethyl 4,5-difluoro-2-isocyano-1,3-dihydroindene-2-carboxylate<[C-]#[N+]C1(C(=O)OCC)Cc2ccc(F)c(F)c2C1>
> C. 3-[3-(cyclohexylamino)-6-methylimidazo[1,2-a]pyridin-2-yl]benzene-1,2-diol<Cc1ccc2nc(c3cccc(c3O)O)c(NC3CCCCC3)n2c1>
> D. 2-[3-(cyclohexylamino)-6-methylimidazo[1,2-a]pyridin-2-yl]-6-ethoxyphenol<CCOc1cccc(c1O)c1c(NC2CCCCC2)n2cc(C)ccc2n1>
> ```
>
> - Step 5: The LM chooses C as the final answer with a confidence level of 7, while the GNN model chooses **A** (the rank 1 among all the molecule corpus). The ground truth is **C**, 3-[3-(cyclohexylamino)-6-methylimidazo[1,2-a]pyridin-2-yl]benzene-1,2-diol.
>
> You make an excellent point that our paper is currently missing detailed case examples illustrating this process. We will address this oversight by adding case studies to the appendix to demonstrate how our model selects the final prediction from the GNN's Top-K candidates. Thank you again for highlighting this gap - the additional examples will greatly improve the clarity and completeness of our approach description.
>
> > **Comment 2: Novelty and Contribution** — "The novelty of methodology looks limited. The main idea contribution lies mostly on the prompt design part."
>
> We appreciate you highlighting this context around the goal and scope of our exploration. The additional background helps situate the motivation for our approach. Recent studies have demonstrated that large language models alone struggle with molecular understanding, hampering performance on tasks like reaction prediction [1]. This paper explores new methods to combine the molecular modeling strengths of graph neural networks (GNNs) with the broad chemical knowledge and reasoning ability of language models (LMs).
>
> Importantly, our focus here is investigating techniques for integrating these models, rather than developing cutting-edge prompts. So the prompting methodology prioritizes intuitive chemical reaction questions answering rather than pursuing state-of-the-art engineering solutions for this task.
>
> > **Comment 3: Randomness of In-context Confidence Scores** — "The confidence score design is also a little bit weird, what’s the intuition to set the score choice as {1,2} and {8,9}? Why is randomly picking up a score reasonable? Why do strict score assignments (e.g., 1 for bad and 9 for good) not work? Missing explanation to this further limits the novelty of the idea itself." "As discussed in weakness, for your confidence score strategy, why pick {1,2} and {8,9}? Why not just {1} and {9} or {1,2,3} and {7,8,9}? Wouldn’t the randomness bias the model?"
>
> Thank you for the valuable comments on our prompt design. Our primary intuition for incorporating randomness is to faithfully mimic human behavior when answering the multiple-choice chemistry question. Typically, humans exhibit varying degrees of confidence, even when answering correctly.
>
> Your comment has been enlightening, suggesting an intriguing question of exploration. What if we design diverse confidence labels for ReLM? In response, we **add additional comparative experiments** with MolR+Vicuna at $K=4$, using both fixed and randomized labels for the in-context examples.
>
> Table 1. Accuracy over fixed and randomized in-context confidence scores.
>
> |                                 | Imidazo | NiCOlit | rexgen-30k | rexgen-40k |
> | ------------------------------- | ------- | ------- | ---------- | ---------- |
> | MolR                            | 51.30%  | 43.70%  | 47.13%     | 44.89%     |
> | Fixed: {1} and {9}              | 91.93%  | 53.94%  | 48.65%     | 45.62%     |
> | Randomized: {1,2} and {8,9}     | 87.76%  | 52.32%  | 49.88%     | 47.27%     |
> | Randomized: {1,2,3} and {7,8,9} | 91.41%  | 53.78%  | 48.51%     | 45.52%     |
>
> We can observe that: using varying confidence scores causes slight fluctuations in experimental outcomes, though the degree varies by dataset. This further demonstrates that the effectiveness of our ReLM lies not in the details of prompt design. Instead, it stems from our main idea of amalgamation of the molecular modeling ability inherent to GNNs with the vast reaction knowledge of the language model.
>
> We will expand upon these investigations in the appendix.
>
> > **Comment 4: Top-$K$ Candidates Selection** — "Eqn (1): What is the size of the reactants R? Is it a list that contains all the reactants or a single reactant? Is it fixed for a query? If there are multiple reactants, how do you leverage GNN to select top-K candidates? Do you pick top-K from only one of the reactants or all of them?"
>
> Thank you for catching the unclear explanation. To clarify how Equations (1) and (2) operate:
> - $R$ refers to the set of multiple query reactants and $r$ denotes an individual reactant
> - $P$ represents multiple product molecules on the right side of the reaction, with $p$ as a single product
> - Equation (1) sums reactant representations to retrieve Top-$K$ product candidates $P_j$
> - Importantly, $P_j$ comes from a predefined corpus so Equation (1) computes the distance between $R$ and all $P_j$
> - The GNN ranks distance $D(R, P_j)$ for all $P_j$ sets and picks the top $K$ closest as the predictions (per equation 2)
>
> In summary, the reactant representations select the most similar Top-$K$ product sets $P_j$ from the corpus based on distance. Thank you again for identifying this point needing further improvement - please let us know if the revised explanation fails to clarify the process.
>
> > **Comment 5: Experiment Metrics and Future Work** — Could you explain the difference between the baseline between the first row (method name) and last row (hit@$K$), and explain why hit@$K$ is not enough to get results? How do you plan to extend this work to increase the generalizability and adaptability of ReLM?
>
> Thank you for your interest in the experimental setup. To summarize:
> - The GNN transforms reaction prediction into a ranking task over a predefined product vocabulary $C$
> - For a given reactant $R$, the GNN ranks all $P\in C$ candidates by distance to $R$ (Equation 1)
> - Accuracy is if the ground truth product is rank 1, i.e., GNN’s performance.
> - Hit@$K$ is if the ground truth product is in the Top-$K$ ranks.
>
> If this is not clear enough, you can imagine that each reaction only produces one product (this holds in most cases), and the goal is to find this product out of the predetermined vocabulary $C$. We will revise the related sections to elucidate the ranking process and the scope of the product candidate space more comprehensively.
>
> Our plan to increase the generalizability and adaptability of this work:
> - The current ReLM approach is limited by the hit@K metric of the GNN model. Going forward, we may seek to address this limitation, such as by coupling the language model and graph neural network at the feature level.
> - Additionally, we aim to incorporate experimental procedure data from wet lab chemistry experiments to refine the reaction conditions in a more step-by-step manner, providing the model with richer textual information during reasoning.
>
>
> Thank you again for the insightful feedback.
>
> [1] Augmenting large-language models with chemistry tools

---

### Official Review · Reviewer_ZhrP · 2023-08-05

**Soundness:** 3

**Excitement:**

3: Ambivalent: It has merits (e.g., it reports state-of-the-art results, the idea is nice), but there are key weaknesses (e.g., it describes incremental work), and it can significantly benefit from another round of revision. However, I won't object to accepting it if my co-reviewers champion it.

**Paper Topic And Main Contributions:**

This paper proposes a in-context learning method that utilized both the language model and GNN methods for the chemical reaction task. First, this method extracts the top-K answer candidates by GNN and similarity measurement. These will be the answer candidates. Second, for the in-context learning, this framework also extracts the top-N nearest neighbors. The similarity is calculated using the embeddings. The proposed method outperforms the SOTA GNN-based method. More information can be added into this framework. For example, you can extract some related text about the molecules or generate the description using the language models.

**Reasons To Accept:**

Strengths:
1.	This paper proposes a new in-context learning framework that combines both the language model and GNN method.
2.	The confidence score strategy is proposed to enhance the robustness and interpretability.

**Reasons To Reject:**

Weaknesses:
1. In the section 3.1, if we consider all molecules or a very large set of molecules as the candidate pool C, it might be hard to select related answer candidates there. In addition, it is not clear about Top-N nearest neighbor selection. Please share more information including how to get the embeddings of reaction samples.
2. This method hasn't considered the structure relations between input molecules and the output prodcuts.

**Reproducibility:**

3: Could reproduce the results with some difficulty. The settings of parameters are underspecified or subjectively determined; the training/evaluation data are not widely available.

**Reviewer Confidence:**

4: Quite sure. I tried to check the important points carefully. It's unlikely, though conceivable, that I missed something that should affect my ratings.

---

> ### Author Rebuttal · Authors · 2023-08-29
>
> We thank the reviewer for the thorough and valuable feedback. To address your concerns, we present the point-to-point responses as follows.
>
> > **Comment 1: Candidate Selection** — "In the section 3.1, if we consider all molecules or a very large set of molecules as the candidate pool C, it might be hard to select related answer candidates there."
>
> Thank you for pointing out the ambiguity in the methodology of this paper. We totally agree with you that it’s hard for language models to tackle a large amount of candidate molecules.
>
> **Additional experiments** were conducted to demonstrate this perspective. By varying the value of $K$ in our method, we test the effect of the number of answer candidates (using Vicuna as the language model and MolR as the GNN), as shown in Table 1. While our model's performance remained relatively stable across different values of $K$, the redundant information introduced by an excessively large $K$ ($K=10$) still led to an obvious accuracy drop. This implies that increasing $K$ to the size of the entire database would result in a significant decrease in model accuracy due to information redundancy.
>
> Table 1. Experiments over different $K$ values.
>
> |           | Imidazo  |             | NiCOlit  |             | rexgen-30k |             | rexgen-40k |             |
> | --------- | -------- | ----------- | -------- | ----------- | ---------- | ----------- | ---------- | ----------- |
> | $K$ value | ReLM Acc | GNN hit@$K$ | ReLM Acc | GNN hit@$K$ | ReLM Acc   | GNN hit@$K$ | ReLM Acc   | GNN hit@$K$ |
> | 1         | -        | 51.30%      | -        | 43.70%      | -          | 47.13%      | -          | 44.89%      |
> | 2         | 79.43%   | 81.25%      | 41.92%   | 59.30%      | 48.40%     | 55.57%      | 44.51%     | 51.10%      |
> | 3         | 91.41%   | 94.53%      | 51.00%   | 64.09%      | 49.75%     | 58.46%      | 47.28%     | 56.25%      |
> | 4         | 87.76%   | 97.92%      | 52.32%   | 67.91%      | 49.88%     | 61.10%      | 47.27%     | 58.69%      |
> | 5         | 90.89%   | 98.70%      | 55.32%   | 72.46%      | 46.70%     | 62.91%      | 44.68%     | 61.05%      |
> | 7         | 89.84%   | 99.48%      | 56.86%   | 79.85%      | 46.66%     | 65.18%      | 44.72%     | 65.37%      |
> | 10        | 85.42%   | 99.74%      | 45.82%   | 85.38%      | 45.33%     | 69.84%      | 42.89%     | 68.39%      |
>
> To address this issue, graph neural networks are utilized to encode molecular graph embeddings and perform candidate ranking to obtain the Top-$K$ most suitable candidates, effectively reducing the number of answer candidates. Importantly, product candidate features can be pre-computed and cached, independent of the query reactants. This allows the graph network to swiftly retrieve candidates over a large candidate pool.
>
>
> > **Comment 2: Top-$N$ Nearest Neighbor Selection** — "In addition, it is not clear about Top-$N$ nearest neighbor selection. Please share more information including how to get the embeddings of reaction samples."
>
> Your concerns regarding the Top-$N$ nearest neighbor selection are well-founded. During the Top-$N$ nearest neighbor selection (i.e., Equation (3)), the objective is to retrieve the $N$ most similar reactions from the training set, given the input set of reactants $R$. This process is implemented by sequentially comparing the summed input reactant features to the summed reactant features for each reaction in the training set. More precisely,, the embedding of the i-th training reaction sample $h_{R_i’}$ is calculated by:
> $$
> h_{R_i’}=\sum_{r\in R_i'} G(r|\theta)
> $$
> where $R_i'$ denotes the reactants of the i-th training sample, and $G(\cdot|\theta)$ is a graph neural network parameterized by $\theta$.
>
> Given the query and training sample embeddings, this paper implements the k-nearest neighbors (KNN) algorithm with cosine similarity as the similarity metric to retrieve the N training samples most similar to the input data.
>
> The reviewer rightly notes that the methodology for this section lacks detailed elaboration. In future revisions, we will expand this portion with a more comprehensive delineation of the procedures.
>
> > **Comment 3: Structure Relations between molecules**  — "This method hasn't considered the structure relations between input molecules and the output products."
>
> Thank you for your thoughtful feedback. We fully concur on the importance of modeling the structural relations between reactants and products for predicting chemical reactions. To enable this, we put forth graph neural networks to encode reactant-product correlations and integrate them with language models to augment reasoning. Owing to this hybrid approach combining GNN molecular understanding and language model knowledge and inference, ReLM demonstrates remarkable gains on certain downstream tasks (as evidenced in Table 1 of the paper).
>
> To further illustrate the aforementioned capabilities, we provide a case study from the Imidazo dataset:
>
> ```
> Question: Which of the following is the major product of the organic reaction between 2,3-dihydroxybenzaldehyde<C(c1cccc(c1O)O)=O>, 5-methylpyridin-2-amine<CC1=CN=C(C=C1)N> and isocyanocyclohexane<[C-]#[N+]C1CCCCC1>, with 2,2,2-trifluoroacetic acid<C(=O)(C(F)(F)F)O> as catalyst?
> A. methyl 4-isocyanocyclohexane-1-carboxylate<[C-]#[N+]C1CCC(C(=O)OC)CC1>
> B. ethyl 4,5-difluoro-2-isocyano-1,3-dihydroindene-2-carboxylate<[C-]#[N+]C1(C(=O)OCC)Cc2ccc(F)c(F)c2C1>
> C. 3-[3-(cyclohexylamino)-6-methylimidazo[1,2-a]pyridin-2-yl]benzene-1,2-diol<Cc1ccc2nc(c3cccc(c3O)O)c(NC3CCCCC3)n2c1>
> D. 2-[3-(cyclohexylamino)-6-methylimidazo[1,2-a]pyridin-2-yl]-6-ethoxyphenol<CCOc1cccc(c1O)c1c(NC2CCCCC2)n2cc(C)ccc2n1>
> ```
>
> Ground truth: C. 3-[3-(cyclohexylamino)-6-methylimidazo[1,2-a]pyridin-2-yl]benzene-1,2-diol
>
> Answer by GPT-3.5 (without the choices):
> ```
> 2-(5-methylpyridin-2-yl)-3-(2,3-dihydroxyphenyl)-1H-pyrrole-1-carbonitrile.
> Confidence Score: 8
> ```
>
> Answer by Vicuna (without the choices):
> ```
> [C-]#[N+]C1CCCCC1>
> Confidence Score: 7
> ```
>
> Answer by Both LM (with the choices):
> ```
> C. 3-[3-(cyclohexylamino)-6-methylimidazo[1,2-a]pyridin-2-yl]benzene-1,2-diol
> Confidence Score: 7
> ```
>
>
> Thank you again for the valuable feedback. Please let us know if any clarification or additional experiments would be helpful.

---

### Official Review · Reviewer_cAqR · 2023-08-12

**Soundness:** 4

**Excitement:**

4: Strong: This paper deepens the understanding of some phenomenon or lowers the barriers to an existing research direction.

**Paper Topic And Main Contributions:**

The paper topic is ReLM, a framework that leverages language models to enhance chemical reaction prediction. The main contributions of the paper are introducing a novel approach that combines language models with graph neural networks to improve the accuracy of chemical reaction prediction, demonstrating the effectiveness of ReLM in out-of-distribution settings.

**Questions For The Authors:**

A. I understand the experimental setup to utilize an out-of-distribution setting to demonstrate the advantages of LLM's knowledge. However, it's also possible to conduct the same experiments in the in-distribution setup. It would be beneficial to include those results in the appendix as well.

B. How about generating confidence scores for each candidate instead of just an overall confidence score for the answer? It would be interesting to compare how these confidence scores differ from the rank produced by GNN. Particularly, in samples where GNN's predict a low rank, but LLM correctly identified the answer, it would be valuable to analyze if such chemicals are far from the distribution in the USPTO dataset.

C. Also, It would be beneficial to represent examples of the reaction type and reaction condition in the Appendix.

**Reasons To Accept:**

A. The paper provides a detailed analysis of the proposed approach, including experiments, statistical analysis, and ablation studies, which can help researchers better understand the strengths and limitations of ReLM.

B. Considering all GNN-based deep neural networks lack the capability to leverage reaction conditions and types, It is noteworthy that ReLM approach aims to utilize the reaction type and reaction condition information in prediction, which are essential factors in chemical reactions. Therefore, this work certainly has aspects that can captivate the readers' interest.

C. Leveraging the knowledge of LLM in out-of-distribution scenarios to improve prediction performance is both consolidated and has a practical impact.

**Reasons To Reject:**

A. The main approach of this work is its use of text information via LLM, which in turn, enables the integration of reaction type and reaction condition. These factors were previously out of reach for traditional GNNs. However, the current experimental setups don't adequately demonstrate how these factors boost the accuracy of chemical reaction predictions. An additional investigation into how the introduction of reaction condition information aids in distinguishing the correct answers from the candidates generated by GNN is needed. It would be illuminating, for instance, to present cases where identical reactants lead to different products under different reaction conditions. Offering such examples could be a valuable way to highlight this.

B. To ensure that the LLM is predicting chemical reactions accurately, the experimental results should incorporate a wider range of values for K, not limited to just 3 and 4. To provide more context, the current framework feeds LLM with answer candidates generated by the GNN. At its core, LLM then selects the correct answer from these candidates. If LLM indeed possesses the capability to accurately pinpoint the correct answer among a variety of candidates, it's important to check whether its performance stays consistent or even gets better as the value of K goes up. Increasing K essentially means adding less plausible options to the candidates, yet it's crucial to see if LLM maintains its accuracy in selecting the correct answer despite the added choices.

C. Referring to the study on LLM's confidence score at **https://arxiv.org/pdf/2205.14334.pdf**, unless supervised fine-tuning (Calibration) is conducted on the confidence score, the score itself is not very reliable. This study(ReLM) claims that merely producing the confidence score increases the accuracy of reaction prediction. However, further analysis on this point is needed. The current baseline for the confidence score strategy is not proper, but other forms of prompts related to reasoning could serve as potential baselines (e.g. **https://arxiv.org/pdf/2205.11916.pdf**).

**Reproducibility:**

4: Could mostly reproduce the results, but there may be some variation because of sample variance or minor variations in their interpretation of the protocol or method.

**Reviewer Confidence:**

5: Positive that my evaluation is correct. I read the paper very carefully and I am very familiar with related work.

**Typos Grammar Style And Presentation Improvements:**

Section 4 Experimental Results, "RQ1 : Can ReLM improve the reasoning ability of graph neural networks in out-of-distribution scenarios?" (222-223)
The term "reasoning ability" needs to be clarified in this sentence. It might confuse the reader into thinking it refers to the reasoning for the interpretability of predictions.

---

> ### Author Rebuttal · Authors · 2023-08-29
>
> Thank you for taking the time to review our submission and provide thoughtful feedback. Your valuable insights into experimental design and model enhancements have greatly inspired our research. In response to the suggestions you have raised, we provide a point-to-point response as follows.
>
> > **Comment 1: Cases of Reaction Conditions** - "The main approach of this work is its use of text information via LLM, which in turn, enables the integration of reaction type and reaction condition. These factors were previously out of reach for traditional GNNs. However, the current experimental setups don't adequately demonstrate how these factors boost the accuracy of chemical reaction predictions. An additional investigation into how the introduction of reaction condition information aids in distinguishing the correct answers from the candidates generated by GNN is needed. It would be illuminating, for instance, to present cases where identical reactants lead to different products under different reaction conditions. Offering such examples could be a valuable way to highlight this."
>
> Thank you for insightful suggestions regarding the importance of demonstrating the influence of reaction condition. Following your suggestion, we add addtional case studies in the Imidazo dataset to illuminate the pivotal role of reaction conditions of ReLM.
>
> For clarity, we present two cases, each showcasing a subtly different prompt design, accompanied by ChatGPT's sketched responses. We intentionally exclude in-context examples, allowing the language model to elucidate its answers.
> ```
> Question1: Which of the following is the major product of the organic reaction between 2,3-dihydroxybenzaldehyde<C(c1cccc(c1O)O)=O>, 5-methylpyridin-2-amine<CC1=CN=C(C=C1)N> and isocyanocyclohexane<[C-]#[N+]C1CCCCC1>, with 2,2,2-trifluoroacetic acid<C(=O)(C(F)(F)F)O> as catalyst? The given reaction is a multicomponent reaction that leads to the synthesis of diverse imidazopyridines.
> A. ethyl 4,5-difluoro-2-isocyano-1,3-dihydroindene-2-carboxylate<[C-]#[N+]C1(C(=O)OCC)Cc2ccc(F)c(F)c2C1>
> B. methyl 4-isocyanocyclohexane-1-carboxylate<[C-]#[N+]C1CCC(C(=O)OC)CC1>
> C. 3-[3-(cyclohexylamino)-6-methylimidazo[1,2-a]pyridin-2-yl]benzene-1,2-diol<Cc1ccc2nc(c3cccc(c3O)O)c(NC3CCCCC3)n2c1>
> D. 2-[3-(cyclohexylamino)-6-methylimidazo[1,2-a]pyridin-2-yl]-6-ethoxyphenol<CCOc1cccc(c1O)c1c(NC2CCCCC2)n2cc(C)ccc2n1>
>
> ChatGPT sketched response:
> The question describes a multi-component reaction between three reactants:
> ….
> Given the reactants, the most logical major product seems to be:
> C. 3-[3-(cyclohexylamino)-6-methylimidazo[1,2-a]pyridin-2-yl]benzene-1,2-diol.
> ```
> ```
> Question2: Which of the following is the major product of the organic reaction between 2,3-dihydroxybenzaldehyde<C(c1cccc(c1O)O)=O>, 5-methylpyridin-2-amine<CC1=CN=C(C=C1)N> and isocyanocyclohexane<[C-]#[N+]C1CCCCC1>?
> A. ethyl 4,5-difluoro-2-isocyano-1,3-dihydroindene-2-carboxylate<[C-]#[N+]C1(C(=O)OCC)Cc2ccc(F)c(F)c2C1>
> B. methyl 4-isocyanocyclohexane-1-carboxylate<[C-]#[N+]C1CCC(C(=O)OC)CC1>
> C. 3-[3-(cyclohexylamino)-6-methylimidazo[1,2-a]pyridin-2-yl]benzene-1,2-diol<Cc1ccc2nc(c3cccc(c3O)O)c(NC3CCCCC3)n2c1>
> D. 2-[3-(cyclohexylamino)-6-methylimidazo[1,2-a]pyridin-2-yl]-6-ethoxyphenol<CCOc1cccc(c1O)c1c(NC2CCCCC2)n2cc(C)ccc2n1>
>
> ChatGPT sketched response:
> The given reactants are:
> …
>
> Drawing out the structure, the major product is:
> O=C(O)C(NC(=O)c1cccc(c1O)O)N=C(N)c2cccnc2C1CCCCC1.
>
> ```
>
> Question 1 is the target reaction given correct catalyst. While question2 omits the important catalyst. For GNN, it incorrectly predict B as the answer, which is not even the correct product without the given catalyst. In contrast, language model accurately identify option C for Question 1, which aligns with the ground truth. Suprisingly, for Question 2, the language model predict a product that was not present in the initial candidate pool.
>
> These case studies clearly demonstrate that identical reactants yield different products under different reaction conditions, further reinforcing our driving hypothesis: the incorporation of reaction type and condition boost the predictive accuracy of ReLM.
>
> We will incorporate such case studies into the appendix section of our paper to facilitate readers' comprehension of the significance of reaction conditions in chemical reactions. We are grateful for your recommendation to include these illustrative examples.
>
> FYI, the detailed analysis of this reaction is provided below:
> This organic reaction involves three components - an aldehyde, an amine, and an isocyanide. Without an added catalyst, the combination of these three reactants undergoes a Passerini reaction, which is a type of multi-component condensation. In a Passerini reaction, the aldehyde gets oxidized in situ to a carboxylic acid. Then, the carboxylic acid, amine, and isocyanide condense together to form an α-acyloxy amide product.
>
> Specifically, in this reaction the aldehyde 2,3-dihydroxybenzaldehyde gets oxidized to 2,3-dihydroxybenzoic acid. This condenses with the amine 5-methylpyridin-2-amine and the isocyanide isocyanocyclohexane. The expected major product is an α-acyloxy amide with the IUPAC name 2-[[amino-(2-cyclohexylpyridin-3-yl)methylidene]amino]-2-[(2,3-dihydroxybenzoyl)amino]acetic acid. The structure contains the 2,3-dihydroxybenzoyl group from the oxidized aldehyde, the 5-methylpyridin-2-amine portion, and the isocyanocyclohexane portion.
>
> However, in the presence of the trifluoroacetic acid catalyst, the reaction proceeds via an imidazopyridine formation mechanism. The aldehyde and the amine first react in a condensation reaction, with loss of water, to form an imine intermediate. The imine then cyclizes via intramolecular nucleophilic attack to form a dihydropyridine intermediate. Dehydrogenation of the dihydropyridine gives an imidazopyridine heterocycle. The isocyanocyclohexane then attacks the imine carbon of the imidazopyridine, displacing the hydroxyl group. This leads to the major product being a 2,3-disubstituted imidazopyridine attached to the cyclohexylamine group. The specific major product is C.  3-[3-(cyclohexylamino)-6-methylimidazo[1,2-a]pyridin-2-yl]benzene-1,2-diol.
>
>
> > **Comment 2: Model accuracy across $K$ values** - "To ensure that the LLM is predicting chemical reactions accurately, the experimental results should incorporate a wider range of values for K, not limited to just 3 and 4. To provide more context, the current framework feeds LLM with answer candidates generated by the GNN. At its core, LLM then selects the correct answer from these candidates. If LLM indeed possesses the capability to accurately pinpoint the correct answer among a variety of candidates, it's important to check whether its performance stays consistent or even gets better as the value of K goes up. Increasing K essentially means adding less plausible options to the candidates, yet it's crucial to see if LLM maintains its accuracy in selecting the correct answer despite the added choices."
>
> We appreciate the reviewer's insightful suggestion to evaluate our model's performance across a wider range of $K$ values. Following your suggestion, we add an additional experiments over different $K$ values on diverse datasets. The findings are presented in Table 1 using MolR as the GNN backbone and Vicuna as the language model.
>
> We observe that: the ReLM’s performance remains relatively stable as we increase $K$ across a wide range of $K$ values (e.g. $K=2\sim 7$). However, with very large $K$ values (e.g. $K>=10$), the performance does start to degrade.
>
> This drop in accuracy with extremely large K values is likely because the candidates become overly saturated with implausible options, making it more difficult for the language model to discern the correct answer. As the reviewer pointed out, increasing K means adding more distracting and unlikely candidates. While the model is robust to these distractors up to a point, at some threshold of implausible options the task does become more challenging.
>
> Table 1. Experiments over different $K$ values.
>
> |           | Imidazo  |           | NiCOlit  |           | rexgen-30k |           | rexgen-40k |           |
> | --------- | -------- | --------- | -------- | --------- | ---------- | --------- | ---------- | --------- |
> | $K$ value | ReLM Acc | GNN hit@K | ReLM Acc | GNN hit@K | ReLM Acc   | GNN hit@K | ReLM Acc   | GNN hit@K |
> | 1         | -        | 51.30%    | -        | 43.70%    | -          | 47.13%    | -          | 44.89%    |
> | 2         | 79.43%   | 81.25%    | 41.92%   | 59.30%    | 48.40%     | 55.57%    | 44.51%     | 51.10%    |
> | 3         | 91.41%   | 94.53%    | 51.00%   | 64.09%    | 49.75%     | 58.46%    | 47.28%     | 56.25%    |
> | 4         | 87.76%   | 97.92%    | 52.32%   | 67.91%    | 49.88%     | 61.10%    | 47.27%     | 58.69%    |
> | 5         | 90.89%   | 98.70%    | 55.32%   | 72.46%    | 46.70%     | 62.91%    | 44.68%     | 61.05%    |
> | 7         | 89.84%   | 99.48%    | 56.86%   | 79.85%    | 46.66%     | 65.18%    | 44.72%     | 65.37%    |
> | 10        | 85.42%   | 99.74%    | 45.82%   | 85.38%    | 45.33%     | 69.84%    | 42.89%     | 68.39%    |
>
> We will highlight this findings in our revised manuscript to fully characterize the model's prediction capabilities. We appreciate the thoughtful feedback.
>
> > **Comment 3: Analysis of confidence score** - "Referring to the study on LLM's confidence score [1], unless supervised fine-tuning (Calibration) is conducted on the confidence score, the score itself is not very reliable. This study(ReLM) claims that merely producing the confidence score increases the accuracy of reaction prediction. However, further analysis on this point is needed. "
>
> We appreciate the reviewer highlighting the significance of confidence score in our paper, an aspect worthing further exploration.
> As delineated in Table 5 of our paper, reactions with higher confidence scores are prone to accurate predictions compared to their low-confidence counterparts. This underscores the language model's capability for self-assessment.
>
> To further address your concerns, we **add additional experiments**  and **statistical significance test** to show that even uncalibrated confidence scores offer valuable insights for ReLM’s introspection.
>
> Our experiments are conducted on Rexgen-40 dataset, using Vicuna + MolR as backbones and our findings are presented in Table 2.
>
>  Table 2. The distribution of confidence scores.
>
> |                                          | 1    | 2     | 3    | 4    | 5    | 6     | 7     | 8     | 9     |
> | ---------------------------------------- | ---- | ----- | ---- | ---- | ---- | ----- | ----- | ----- | ----- |
> | Distribution over all samples            | 3.2% | 4.3%  | 0.5% | 1.9% | 1.1% | 12.9% | 15.2% | 25.0% | 35.8% |
> | Distribution over correct ReLM choices   | 1.6% | 2.4%  | 0.4% | 1.5% | 1.0% | 10.8% | 16.0% | 27.3% | 39.0% |
> | Distribution over incorrect ReLM choices | 9.6% | 12.1% | 1.0% | 3.3% | 1.5% | 21.3% | 12.1% | 15.9% | 23.2% |
>
> We observe that:
> - There is a clear difference between verbalized probability [1] and confidence score of ReLM. The scores of our method are distributed across 1 to 9, while verbalized probability mainly fall within the 80\% to 100\% range, as tested by Xiong et al. [3]. This result underscores that verbalized probabilities may not serve as reliable indicators for assessing the outcomes of language models. A promising research direction for improved assessment could be the confidence scores inherently generated by the large language model.
> - Confidence score of ReLM exhibites significantly different distributions between correct and incorrect answers, a trend corroborated by a p-value of less than 5%. For incorrect ReLM choices, the proportion of high confidence scores ($7\sim 9$) decreased from 82.3% to 51.2%, while that of low confidence scores ($1\sim 3$) increased from 4.4% to 22.7%.
>
>
> The reviewer makes a fair critique, we fully agree that more evidence is needed before concluding confidence elicitation directly boosts prediction accuracy. In future revisions, we will thoroughly analyze the relationships between confidence scores and performance through controlled experiments.
>
>
> > **Comment 4: Comparing with more Baselines** - "The current baseline for the confidence score strategy is not proper, but other forms of prompts related to reasoning could serve as potential baselines (e.g. [2])."
>
> We appreciate the reviewer suggesting the chain-of-thought (CoT) [2] as new baselines.
>
> Following your suggestion, we **add additional experiments** using Zero-shot, Few-shot CoT, Zero-Shot CoT as new baselines as shown in Table 3.
>
> Table 3. The comparison with more baselines.
>
> |               | Imidazo | NiCOlit | rexgen-30k | rexgen-40k |
> | ------------- | ------- | ------- | ---------- | ---------- |
> | MolR          | 51.30%  | 43.70%  | 47.13%     | 44.89%     |
> | Zero-shot     | 86.98%  | 44.60%  | 30.05%     | 28.62%     |
> | Few-shot CoT | 78.13%  | 30.54%  | 26.70%     | 24.43%     |
> | Zero-shot CoT | 84.38%  | 41.51%  | 31.75%     | 30.17%     |
> | **ReLM**      | 87.76%  | 52.32%  | 49.88%     | 47.27%     |
> | MolR hit@K    | 97.92%  | 67.91%  | 61.10%     | 58.69%     |
>
>
> We try our best to implement task-specific versions of these baseline methods for chemical reaction prediction. However, due to time constraints, we do not explore optimal prompt engineering for these baselines, which we leave for future work.
>
> We observe that all prompt designs offer some benefits. However, our original confidence scoring approach still outperforms these baselines. Specifically, as shown in the case studies below, the language model's CoT analysis may not provide additional insightful information regarding the reaction mechanism. At this stage, incorporating additional analytical steps may introduce more molecular structures, exacerbating the language model's comprehension difficulties. Further step-by-step elucidation of the reaction process likely necessitates incorporating more domain knowledge of chemical reactions.
>
> To provide more detailed insight, we include several case studies that compare our confidence score strategy against reasoning-based baselines.
>
> ```
> Question: Which of the following is the major product of the organic reaction between 2-methoxynaphthalene<COc1ccc2ccccc2c1> and lithium;1,3,5-trimethylbenzene-6-ide<Cc1cc(C)c([Li])c(C)c1>? The given reaction is a Nickel-catalyzed cross-couplings reaction, which forms carbon-carbon (C-C) and carbon-heteroatom (C-X, where X can be O, N, S, etc.) bonds.
> A. 2-(2,4,6-trimethylphenyl)naphthalene<Cc3cc(C)c(c2ccc1ccccc1c2)c(C)c3>
> B. 1-(4-methoxyphenyl)-3,5-dimethylbenzene<COc2ccc(c1cc(C)cc(C)c1)cc2>
> C. 1-(2,4,6-trimethylphenyl)naphthalene<Cc3cc(C)c(c1cccc2ccccc12)c(C)c3>
> D. 1,3,5-trimethyl-2-phenylbenzene<Cc2cc(C)c(c1ccccc1)c(C)c2>
> ```
> Ground truth: A. 2-(2,4,6-trimethylphenyl)naphthalene<Cc3cc(C)c(c2ccc1ccccc1c2)c(C)c3>
>
> Answer by Vicuna under **confidence score strategy**:
> ```
> Answer: A
> Confidence Score: 8
> ```
>
> Answer by Vicuna under **Zero-shot CoT** strategy [2]:
> ```
> Answer: Let's think step by step.
> The organic reaction between 2-methoxynaphthalene< COc1ccc2ccccc2c1> and lithium;1,3,5-trimethylbenzene-6-ide< Cc1cc(C)c([Li])c(C)c1> is a Nickel-catalyzed cross-couplings reaction, which forms carbon-carbon (C-C) and carbon-heteroatom (C-X, where X can be O, N, S, etc.) bonds.
>  The major product of this reaction is 1,3,5-trimethyl-2-phenylbenzene<Cc2cc(C)c(c1ccccc1)c(C)c2>.
>  The other options are 2-(2,4,6-trimethylphenyl)naphthalene<Cc3cc(C)c(c2ccc1ccccc1c2)c(C)c3>, 1-(4-methoxyphenyl)-3,5-dimethylbenzene<COc2ccc(c1cc(C)cc(C)c1)cc2>, and 1-(2,4,6-trimethylphenyl)naphthalene<Cc3cc(C)c(c1cccc2cccc12)c(C)c3>.
>  So, the correct answer is D. 1,3,5-trimethyl-2-phenylbenzene< Cc2cc(C)c(c1ccccc1)c(C)c2>.
> Therefore, the answer (one capital letter, presenting your choice) is
> D. 1,3,5-trimethyl-2-phenylbenzene<Cc2cc(C)c(c1ccccc1)c(C)c2>.
> ```
>
> Answer by Vicuna under **Few-shot CoT** strategy:
> ```
> The reaction between 2-methoxynaphthalene< COc1ccc2ccccc2c1> and lithium;1,3,5-trimethylbenzene-6-ide< Cc1cc(C)c([Li])c(C)c1> is a Nickel-catalyzed cross-couplings reaction, which forms carbon-carbon (C-C) and carbon-heteroatom (C-X, where X can be O, N, S, etc.) bonds.
>  The reaction is a cycloaddition reaction, where the -OH group of lithium is replaced by the oxygen atom from the benzene moiety. The result is the attachment of the naphthalene ring to the benzoic acid moiety through an oxygen atom, yielding the compound 2-(2,4,6-trimethylphenyl)naphthalene. This process involves a substitution reaction, replacing the iodine atom with the oxygen atom from the alcohol group.
>  Option D is duplicated and should not be considered. This prediction is based on the cycloaddition reaction and the potential for rearrangements during the reaction in aqueous conditions.
>  My choice: D
> ```
> In the revised manuscript, we will include detailed analyses highlighting the performance comparison between our confidence scoring approach and these reasoning-based baselines.
>
>
> > **Comment 5: In-distribution Experiments** - "I understand the experimental setup to utilize an out-of-distribution setting to demonstrate the advantages of LLM's knowledge. However, it's also possible to conduct the same experiments in the in-distribution setup. It would be beneficial to include those results in the appendix as well."
>
> Thank you for raising this insightful point about including in-distribution experiments. We agree that showing performance on both in-distribution and out-of-distribution test sets can provide a more comprehensive view of the model's capabilities. Following your suggestion, we **conduct new in-distribution experiments** on USPTO-479 dataset, results shown in Table 4.
>
>
> Table 4. Accuracy on in-distribution settings.
>
> |               | MolR   | LocalRetro |
> | ------------- | ------ | ---------- |
> | GNN           | 88.20% | 56.63%     |
> | **ReLM (Vicuna)** | 87.09% | 62.73%     |
> | GNN hit@4     | 95.27% | 75.83%     |
>
> We observe that: due to the GNN itself achieving relatively high accuracy in in-distribution scenarios, the performance gains afforded by ReLM were limited in those cases. Moreover, the absence of reaction type and condition information in the USPTO dataset posed inference challenges for the language model. Nonetheless, ReLM still exhibited a certain capacity to select the correct answers from the options, without exhibiting significant performance deterioration.
>
> The USPTO dataset comprises numerous chemical reactions. Due to time constraints, we have thus far only conducted experiments under certain settings (Vicuna as the language model, $K=4$). Comprehensive experiments will be completed and reported in future revisions of the paper.
>
> > **Comment 6: Generate Confidence Scores for each candidate** - "How about generating confidence scores for each candidate instead of just an overall confidence score for the answer? It would be interesting to compare how these confidence scores differ from the rank produced by GNN. Particularly, in samples where GNN's predict a low rank, but LLM correctly identified the answer, it would be valuable to analyze if such chemicals are far from the distribution in the USPTO dataset."
>
> Thank you for your insightful comments. I have carefully considered your suggestion to generate confidence scores for each candidate instead of just an overall confidence score for the answer. Generating confidence scores for each candidate might be better aligned with the framework of our proposed methodology. Thanks again.
>
> Following your suggestion, we conducted **new experiments** using MolR as the GNN backbone, Vicuna as the language model, and $K=4$, and recorded the results in Table 5. In addition to model accuracy, we also reported the correlation between the language model and GNN model answers. We used Spearman's rank correlation coefficient as the metric to measure how the rank generated by the language model (calculated based on the option-wise confidence scores) correlates with the original ranking from the GNN model.
>
> Table 5. Results of 'Confidence scores for each candidate' stratety.
>
> |                                           | Imidazo | NiCOlit | rexgen-30k | rexgen-40k |
> | ----------------------------------------- | ------- | ------- | ---------- | ---------- |
> | GNN Acc                                   | 51.30%  | 43.70%  | 47.13%     | 44.89%     |
> | **ReLLM Acc**                                 | 86.98%  | 50.69%  | 44.91%     | 42.10%     |
> | rank correlation over all samples         | 0.3632  | 0.3593  | 0.3457     | 0.3466     |
> | rank correlation over correct GNN choices | 0.4691  | 0.4292  | 0.3883     | 0.3971     |
> | rank correlation over incorrect GNN choices | 0.2425  | 0.2271  | 0.1993     | 0.1743     |
>
>
> We obseve that: the language model's overall rankings positively correlate with the GNN model's rankings, meaning the language model agrees with most, but not all, of the GNN's analytical results. Importantly, for questions where the GNN made incorrect choices (last row in table), the language model's rankings showed a significantly lower correlation with the GNN's rankings. On the contrary, For those where the GNN made correct choices, the correlation between the LLM's answers and the GNN's increased. The above observations indicate that the language model forms its own informed opinions on the answers, and differs from the GNN when GNN makes incorrect choices.
>
>
> > **Comment 7: Examples of Reaction Conditions** - "Also, It would be beneficial to represent examples of the reaction type and reaction condition in the Appendix."
>
> Thank you for highlighting the omission in our paper regarding the description of reaction information. We will systematically arrange the reaction type and reaction condition data from the downstream datasets, and provide its distribution along with illustrative examples in the forthcoming revisions of the paper.
>
> > **Minor Comment 8: Research Question Clarification** - "Section 4 Experimental Results, 'RQ1 : Can ReLM improve the reasoning ability of graph neural networks in out-of-distribution scenarios?' (222-223) The term 'reasoning ability' needs to be clarified in this sentence. It might confuse the reader into thinking it refers to the reasoning for the interpretability of predictions."
>
> We appreciate the reviewer noting the ambiguity of "reasoning ability". To clarify, we will revise the sentence to: "RQ1: Can ReLM improve the reaction prediction capability of graph neural networks on out-of-distribution data?" This refers specifically to prediction accuracy, not interpretability. Please let us know if this revision provides sufficient clarification. Thank you for improving the clarity of our work.
>
> [1] Teaching models to express their uncertainty in words
>
> [2] Large Language Models are Zero-Shot Reasoners
>
> [3] Can LLMs Express Their Uncertainty? An Empirical Evaluation of Confidence Elicitation in LLMs

---

### Meta-Review · Area_Chair_AZDx · 2023-09-18

**Recommendation:** 4

**Metareview:**

This paper proposes ReLM, a framework for predicting chemical reactions. ReLM combines the capability of language models as well as graph neural networks to provide answer candidates, followed by the confidence score strategy. All reviewers agreed on the well-motivated approach for combining LM and GNN, as well as postiive experiment results.

---

### Decision · Program_Chairs · 2023-10-07

**Decision:**

Accept-Findings

**Comment:**

This paper proposes ReLM, a framework for predicting chemical reactions. ReLM combines the capability of language models as well as graph neural networks to provide answer candidates, followed by the confidence score strategy. All reviewers agreed on the well-motivated approach for combining LM and GNN, as well as postiive experiment results.